



# Numerical Analysis of Transonic Flow over the FFA-W3-211 Wind Turbine Tip Airfoil

Maria Cristina Vitulano[1,2], Delphine De Tavernier[2], Giuliano De Stefano[1], and Dominic von Terzi[2]

[1]Engineering Department, University of Campania Luigi Vanvitelli, 81031 Aversa, Italy
[2]Aerospace Engineering Faculty, Delft University of Technology, 2629HS Delft, The Netherlands

**Correspondence:** Maria Cristina Vitulano (mariacristina.vitulano@unicampania.it)

**Abstract.** Modern wind turbines are the largest rotating machines ever built, with blade lengths exceeding one hundred meters. Previous studies demonstrated how the flow around the tip airfoils of such large machines reaches local flow Mach numbers (Ma) at which the incompressibility assumption might be violated, and, even in normal operating conditions, local supersonic flow could appear. In the present study, a numerical analysis of the FFA-W3-211 wind turbine tip airfoil is performed. The results are obtained by means of the application of numerical tools: (1) Xfoil with the Prandtl-Glauert compressible correction and (2) Computational Fluid Dynamic (CFD) simulations, where an Unsteady Reynolds-Averaged Navier-Stokes (URANS) model is used. A preliminary validation of the latter CFD model is performed to demonstrate that the URANS approach is a viable method for predicting the aerodynamic performances in compressible and transonic flow that provides additional and more reliable information compared to the classical compressibility corrections. From this study, three key findings can be highlighted. Primarily, the main transonic features of the FFA-W3-211 wind turbine tip airfoil have been assessed, selecting specific test cases of particular industrial interest. Then, the threshold between subsonic and supersonic flow is provided, considering also an increase of the Reynolds number (Re) from a characteristic value used in the wind tunnel experiments to the one realistic for large rotors. A strong dependence on this quantity is observed, revealing that, for the same Mach number, also the Reynolds number plays a crucial role in promoting the occurrence of transonic flow. Finally, the possible presence or absence of shock waves was investigated. The results indicate that the appearance of transonic flow is a necessary but not a sufficient condition to lead to shock formation.

## 1 Introduction

According to the Global Wind Energy Council (GWEC), the global demand for wind energy recently reached an annual growth rate of 19%, increasing from 4.7 GW in 1995 to 651 GW in 2019 (GWPC , 2019). However, this number is expected to rise further in order to limit the global warming temperature increment by 2050 to 1.5°C (Hoegh-Guldberg et al. , 2019). Concurrently, since the 2000s, the wind turbine rotors have significantly grown in size, exceeding the blade length of 100m. Current and next-generation (large) wind turbines exhibit high relative velocities, approaching 100m/s at the blade tip. In these operational conditions, wind turbine blades encounter air flows at increased local flow Mach number, where the usual





incompressibility assumption may be violated, and even local supersonic flow could appear at the outboard blade sections. Accordingly, the aerodynamic characterization of compressible and transonic flow assumes a crucial role in designing next-generation wind turbines, as well as in assessing risks regarding performance, loading and fatigue.

In the relevant literature, various efforts have recently been made to identify the effect of high tip speeds on the aerodynamic features of large rotors. For instance, within the EU project AVATAR (AdVanced Aerodynamic Tools of lArge Rotors), wind
tunnel tests and CFD calculations were used to assess design tools for large wind turbines (Ceyhan et al. , 2016). As part of this research, Sørensen et al. (2018) conducted a 2D study on the impact of compressibility under the operational conditions experienced at the tip of a 180m rotor. This study confirmed the possibility that the blade tip might achieve high relative velocities, approaching thirty percent of the speed of sound. It was also shown that simple compressibility corrections are insufficient outside the linear region of the lift curve, where a full compressible formulation should be applied. Yan and
Archer (2018) investigated the compressibility effects on the performance of large horizontal axis wind turbines (HAWT), highlighting that neglecting them for high inflow Mach numbers ($M_\infty > 0.5$) could result in an overestimation of wind farm power production. Campobasso et al. (2018) and Ortolani et al. (2020) studied the compressible aerodynamics of a floating offshore wind turbine, revealing that the compressibility effects produce an increase in the peak rotor power. More recently, Cao et al. (2023) compared aerodynamic performances of the IEA-15MW offshore reference wind turbine (RWT), defined
by Gaertner et al. (2020), under incompressible and compressible conditions. Their analysis shows that the compressible flow simulation overestimated the thrust value by $1.4\%$ against the incompressible simulation, with the predicted torque being nearly $11\%$ higher. A similar analysis of the compressible aerodynamics of the IEA-15MW RWT was carried out by Mezzacapo et al. (2023) employing the Unsteady Reynolds-Averaged Navier-Stokes (URANS) approach. This work confirmed that URANS calculations can provide a preliminary insight into the aerodynamic behaviour of wind turbine blades in the compressible
regime.

While previous studies acknowledged the potential compressibility effects for large wind turbines due to the air density variations, it is only very recently that De Tavernier and von Terzi (2022) showed how next-generation machines might suffer from local supersonic flow, which may lead to severe lifetime degradation. They analyzed the operating conditions of the IEA-15MW RWT using OpenFAST and employing compressibility corrections for the airfoil polars. This study reveals that, near
the cut-out wind speed, supersonic Mach numbers can locally appear on a portion of the tip airfoils. In such conditions, the blade is operating at large negative angles of attack, and the flow is significantly accelerated over the suction side of the blade. Additionally, in off-design conditions, transonic flow may occur even at lower wind speeds. For all cases, the inflow turbulence was identified as the driving factor, which requires an unsteady analysis. Recently, these results were validated experimentally by Aditya et al. (2024). In these experiments, also supersonic flow was identified without a clear presence of shocks.

De Tavernier and von Terzi (2022) showed the location and timings of transonic flow occurrence, but some uncertainty exists, due to the use of compressibility corrections, as it was previously shown by Sørensen et al. (2018). Moreover, the method cannot predict the flow behaviour once the transonic flow has developed and, therefore, if shocks and associated effects like buffeting actually occur.





The behaviour of wind turbine airfoils under transonic flow conditions remains still uncertain. For instance, Hossain et al.
(2013) investigated the propagation of shocks on the NREL Phase VI S809 airfoil by means of URANS calculations. However, they considered an inflow Mach number of 0.8, which is very far from what wind turbines normally encounter. Moreover, they only focused on positive angles of attack, while a typical HAWT commonly operates at negative incidences at the high wind speeds needed for transonic flow to occur. Nevertheless, if transonic flow were to appear, shocks, flow separation, and buffeting could be expected. These phenomena, in turn, may give rise to vibrations that could significantly compromise the structural
integrity and overall lifespan of the turbine, posing potential challenges to its long-term performance and reliability.

The purpose of this paper is to investigate the aerodynamic features of the FFA-W3-211 wind turbine tip airfoil under the above-discussed flow conditions. The present analysis provides further evidence on the appearance of local supersonic flow and shock waves for wind turbine tip airfoils of modern large turbines and identifies the combinations of Mach number and angles of attack for which these conditions can occur.

The fluid turbulence is modelled by means of the URANS approach, where the resolved mean flow is assumed two-dimensional. The numerical model is first validated at low Mach number by comparing the numerical results against reference experimental and numerical data provided by Bertagnolio et al. (2001). The proposed method is applied at higher Mach numbers, by adjusting the fluid viscosity to keep the Reynolds number constant. Then, the effect of increasing the Reynolds number is examined. Moreover, the threshold between subsonic and supersonic flow is found in terms of inflow Mach number
and angle of attack, by also performing a comparison between URANS and compressibility correction approaches. Finally, the possible presence of shock waves under transonic flow conditions is analyzed.

The rest of this manuscript is organized as follows. In Section 2, the CFD methodology is introduced, presenting the particular case study. In the present section, the main numerical settings are discussed, and the validation of the computational model is provided. The results of the present simulations are given in Section 3, where three key points can be identified. First,
the transonic features of the wind turbine tip airfoil are highlighted for specific test cases of particular industrial interest. Then, the threshold between supersonic and subsonic regimes is identified. Finally, the possible presence of shock waves is detected. Some concluding remarks are offered in Section 4.

## 2 Computational model

### 2.1 Case study

The present investigation is focused on a CFD analysis of the FFA-W3-211 airfoil, that is employed at the tip of the 240m-diameter rotor of the IEA-15MW RWT. This turbine has been designed (with input from industry) to be relevant for the next-generation wind turbines that are currently being built or designed, representing the state-of-the-art in the public domain (Gaertner et al. , 2020). Note that the same airfoil will be used for its announced successor, with 280m-diameter rotor and 22MW power, which is currently under development.





A recent analysis, carried out by De Tavernier and von Terzi (2022) by means of Xfoil and the open-source wind tur-
bine analysis tool OpenFAST, revealed that, for this particular wind turbine, transonic flow could occur in normal operating
conditions, especially at cut-out wind speeds, when the blade is pitched to large negative angles of attack for power shedding.

  Figure 1 shows the corresponding operational conditions, in terms of combinations of the angle of attack and inflow Mach
number, for which transonic flow appears. In this picture, each dot represents a distinct time instant of the real operational
condition of the IEA-15MW RWT at $97\%$ of the blade radius, for an inflow velocity of 25m/s and a Reynolds number of $10^7$.

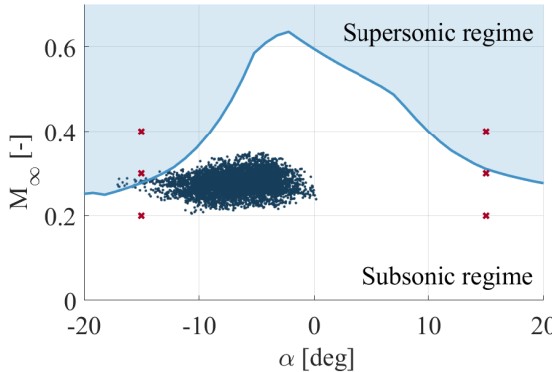

**Figure 1.** Operational conditions of the IEA-15MW RWT at $97\%$ of the blade radius (De Tavernier and von Terzi , 2022). The red crosses
represent some of the particular cases considered here

.

  The aim of the first part of this study is to prove that transonic flow can appear even in normal operation conditions, as
predicted by De Tavernier and von Terzi (2022). To this end, a set of test cases of industrial interest is selected, as illustrated
by the red crosses in Figure 1. In particular, for this first part, two different angles of attack are analyzed, $\alpha = -15°$ and $15°$, as
is illustrated in the figure. These specified values are empirically regarded as critical operating conditions for this wind turbine.
Once the appearance of transonic flow is demonstrated, the threshold between the supersonic and subsonic regime (shown
by the blue curve in Figure 1) is reproduced using URANS computations, for Re = $1.8 \times 10^6$, $3.5 \times 10^6$, $9 \times 10^6$.

### 2.2 Numerical settings

The present numerical analysis is performed by following the URANS approach. The mean compressible flow governing
equations, which are not reported here for brevity, can be found in Willcox (2006).
All simulations are conducted by means of the open-source CFD software OpenFoam, using the solver *rhoPimpleFoam*,
which is suitable for transient, turbulent, and compressible flows (OpenFoam Foundation , 2016). Fully turbulent flow con-
ditions are considered while employing the two-equation $k$-$\omega$ Shear Stress Transport (SST) eddy-viscosity turbulence model
(Menter , 1994). Free-stream boundary conditions are used for pressure, velocity and temperature fields, while no-slip boundary
conditions are enforced at the solid surface. Also, to save computational resources, wall functions are used to model the bound-
ary layer region. A second-order discretization is imposed on the spatial variables, with the first-order implicit Euler scheme





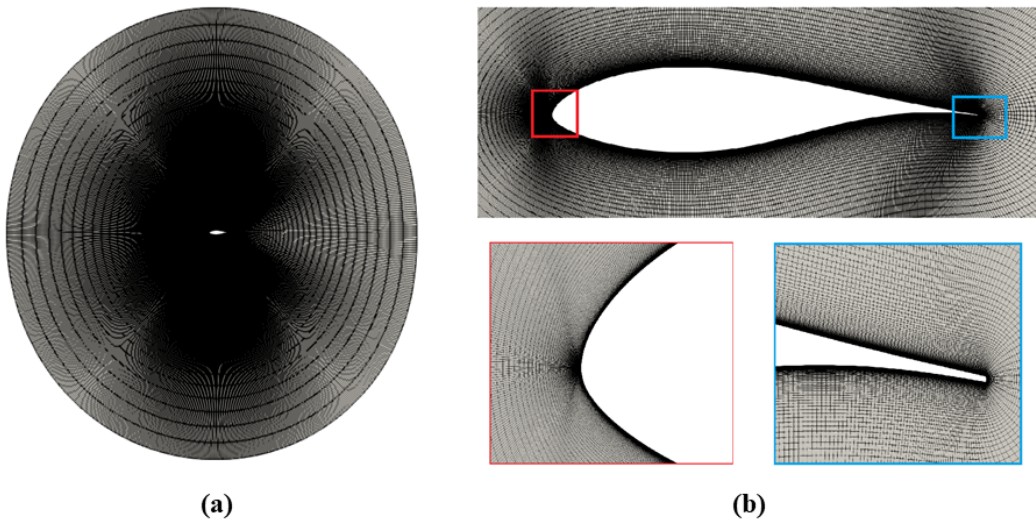

**(a)**      **(b)**

**Figure 2.** Computational domain (a) and spatial mesh, with zoomed views of the airfoil (b) top, the leading and trailing edge regions (b) bottom.

being employed for the time integration. The latter is performed with a variable time step, where the maximum Courant number is fixed at the value of 0.5 to ensure temporal accuracy while maintaining numerical stability. An inflow turbulent intensity of 0.15% is imposed, following similar investigations performed by Bertagnolio et al. (2001).

In this work, a two-dimensional computational domain is considered for resolving the mean turbulent flow around the
airfoil. The domain is discretized by means of a structured computational mesh that is generated with the open-source software Construct2D (Prosser , 2014), using an O-grid topology. Figure 2 shows the circular computational domain, with the external boundary located at ten chords from the origin. The details of the mesh around the airfoil, leading and trailing edges are highlighted in the same figure. To assess the effect of the domain size upon the numerical solution, the pressure coefficient distribution for a particular flow configuration is considered, at the angle of attack $\alpha = 7.99°$, and the flow Reynolds number
Re$= 1.8 \times 10^6$. The solutions corresponding to three different domain radii, that are 10 (Domain I), 30 (Domain II), and 50 (Domain III) chord lengths, are shown in Figure 3, compared to the reference experimental and numerical data in Bertagnolio et al. (2001). Note that the discretization of the airfoil boundary, consisting of $N = 499$ points, is maintained constant for the three different cases. Assuming the largest domain as the reference one, the mean error associated with the other two domains is evaluated as:

$$Err^{I,II} = \frac{1}{N} \sum_{i=1}^{N} \frac{|C_{p,i}^{I,II} - C_{p,i}^{III}|}{|C_{p,i}^{III}|} \ .$$              (1)

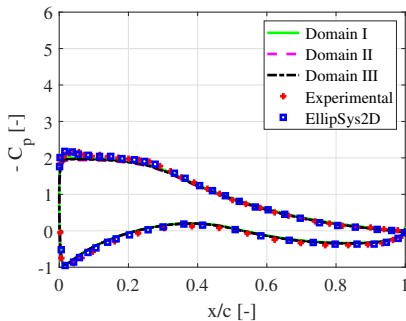

**Figure 3.** Mean pressure coefficient distribution, for varying domain size, at $\alpha = 7.99°$ and Re $= 1.8 \times 10^6$, compared to reference data (Bertagnolio et al. , 2001).

Based on the results summarized in Table 1, the radius of 10 chords, which was also recently used in a similar study by Carta et al. (2022), can be considered as representing the best compromise between the accuracy of the solution and the computational cost.

**Table 1.** Mean error per node for varying domain size.

| Mesh | Error |
| --- | --- |
| Domain I | $4.3\,\mathrm{e}^{-3}$ |
| Domain II | $7.7\,\mathrm{e}^{-5}$ |

Determined the domain size, the proposed CFD model is validated by comparing the pressure distribution at Re $= 1.8 \times 10^6$,

and three different incidences that are $\alpha = 7.99°, 14.98°$, and $20.33°$, against the numerical and experimental data reported in Bertagnolio et al. (2001). In fact, by inspection of Figure 4, where the URANS solutions for three different mesh resolutions are reported, there exists a certain deviation on the suction side of the airfoil, corresponding to the separated flow region, for high angles of attack. This could be attributable to differences in the dynamic stall behaviour between the numerical simulations and the experiments. It is important to highlight that it is actually unclear how the airfoil is tripped within the reference experiments.

Furthermore, the EllipSys2D simulations were carried out for steady flow conditions, while the present unsteady results are averaged in time.

Once the numerical set-up has been validated, the effect of increasing the Mach number is investigated, changing the free-stream velocity magnitude by adjusting the fluid viscosity in order to keep the Reynolds number constant. Furthermore, also the effect of increasing the Reynolds number is analyzed, modifying both the velocity and the fluid viscosity.





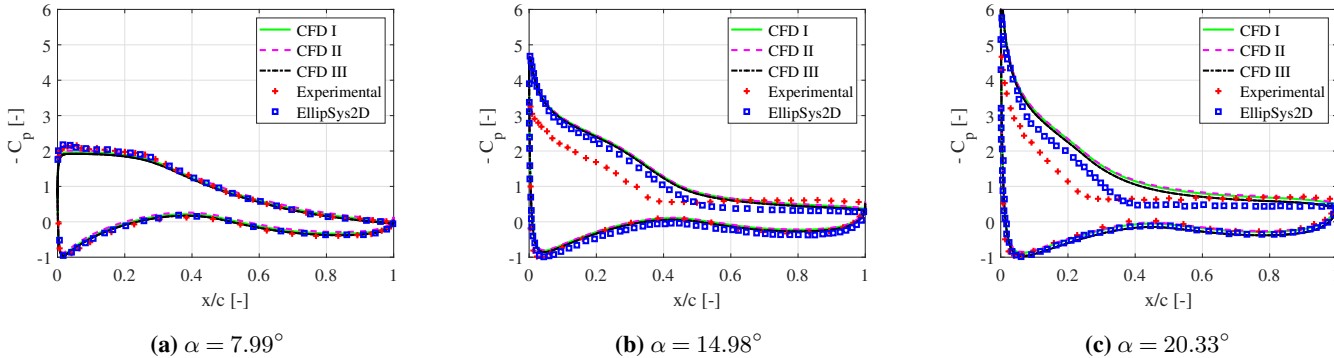

**(a)** $\alpha = 7.99°$ | **(b)** $\alpha = 14.98°$ | **(c)** $\alpha = 20.33°$

**Figure 4.** Mean pressure coefficient distribution at $\mathrm{Re} = 1.8 \times 10^6$: CFD I ($7.4 \times 10^4$ cells), CFD II ($10 \times 10^4$ cells), and CFD III ($12 \times 10^4$ cells), compared to reference data (Bertagnolio et al. , 2001).

### 2.3 Transonic flow

In this study, the threshold for which transonic flow occurs has been established by comparing URANS data with results obtained by simple compressibility corrections. Fixing the inflow Mach number, the minimum pressure coefficient is evaluated for $n$ different angles of attack, obtaining an array of discrete values as $[(C_{p_{\min,1}}, \alpha_1), \ldots, (C_{p_{\min,n}}, \alpha_n)]$. These points are interpolated to determine a continuous law that relates the minimum pressure peak and the angle of attack. Finally, the pressure suction peak is compared to the critical value at which the flow over the airfoil is locally attaining a supersonic flow, expressed as:

$$C_{p_{\mathrm{crit}}} = \frac{2}{\gamma M_\infty^2} \cdot \left( \left[ \frac{1 + \frac{1}{2}(\gamma - 1)M_\infty^2}{1 + \frac{1}{2}(\gamma - 1)M_{crit}^2} \right]^{\frac{\gamma}{\gamma - 1}} - 1 \right). \tag{2}$$

The angle of attack corresponding to the conditions for which the suction peak reaches the critical value is defined as the critical angle of attack for that specific inflow Mach number.

To assess the appropriateness of this procedure, the comparison with data obtained by means of a compressible correction is performed. The suction peak for a fixed angle of attack is calculated using the software Xfoil (Drela , 2001) in the incompressible regime. This value is adapted as a function of the Mach number using the following Prandtl-Glauert compressibility correction:

$$C_{p_{\mathrm{c}}} = \frac{C_{p_{\mathrm{i}}}}{\sqrt{1 - M_\infty^2}}. \tag{3}$$

This way, the inflow Mach number according to which Equations ( 2) and ( 3) provide the same value, corresponds to the critical Mach number for the fixed angle of attack. Finally, the presence of shock waves is detected by means of the method proposed by Lovely and Hsimes (1999), based on the analysis of the normal Mach number:

$$M_n = \frac{V \cdot \nabla p}{a|\nabla p|} = 1, \tag{4}$$

where $a$ stands for the local speed-of-sound.





## 3 Results

In the following section, three key points are highlighted. First, a set of test cases of industrial interest are selected to prove the appearance of local supersonic flow, even in normal operational conditions, and analyze the main characteristics of transonic flow over the airfoil. Then, the threshold between the supersonic and the subsonic curves is provided, to identify for which incidences the transonic flow occurs, varying the Mach number and the Reynolds number. Finally, the possible presence of shock waves is investigated.

### 3.1 Transonic features over the airfoil

The results of numerical simulations, performed at two different Reynolds numbers, namely Re $= 1.8 \times 10^6$ and $9 \times 10^6$, are presented through qualitative and quantitative analyses. The different computations are performed at three inflow Mach numbers, specifically $M_\infty = 0.2, 0.3$, and $0.4$, analyzing two incidences that are $\alpha = -15°$ and $15°$ , as is summarized in Figure 1.

#### 3.1.1 Qualitative analysis

A strongly compressible flow is established for all the inflow Mach numbers considered here. This is demonstrated in Figure 5 and Figure 6, where the instantaneous contour maps of the local flow Mach number around the airfoil are shown.

According to this preliminary qualitative analysis, transonic flow conditions occur in the case of $M_\infty = 0.4$, for both positive and negative incidences, confirming previous research findings (De Tavernier and von Terzi , 2022). In particular, supersonic flow pockets are visible in Figures 5 (e) and 5 (f), as they are determined by the iso-line of Ma= 0.99.

By inspection of Figure 6, increasing the Reynolds number leads to an earlier appearance of transonic flow, already at $M_\infty = 0.3$, but only for the negative incidence. A small supersonic flow pocket appears close to the leading edge, as is illustrated in Figure 6 (d). The flow field region with local supersonic Mach number increases with the free-stream velocity, as shown, in particular, in Figure 6 (e) and 6 (f).

Figures 7 and 8 show the instantaneous numerical Schlieren images, demonstrating how the presence of possible shock waves characterizes the transonic regime. At the lower Reynolds number that is Re $= 1.8 \times 10^6$, the examination of the density gradient field shows the presence of discontinuities with increasing the inflow Mach number. However, that happens only for the negative incidence, while no discontinuity appears for the positive angle of attack. In fact, this is consistent with the field illustrated in Figure 5, showing a supersonic flow pocket with a very smooth shape for $\alpha = 15°$, which means that the flow field acceleration in so gradual that it doesn't lead to the appearance of a shock wave. At the higher Reynolds number of Re $= 9 \times 10^6$, the effect of compressibility becomes more evident. In fact, the local Mach number attains values as high as 1.36 for the positive incidence, as illustrated in Figure 6 (e). In this case, the local velocity is high enough for a density gradient discontinuity to arise, as shown in Figure 8 (e), but the pressure drop is not sufficient to lead to a subsonic flow regime downstream of the shock wave, resulting in a second supersonic pocket to appear (see Figure 6 (e)). At the negative incidence,

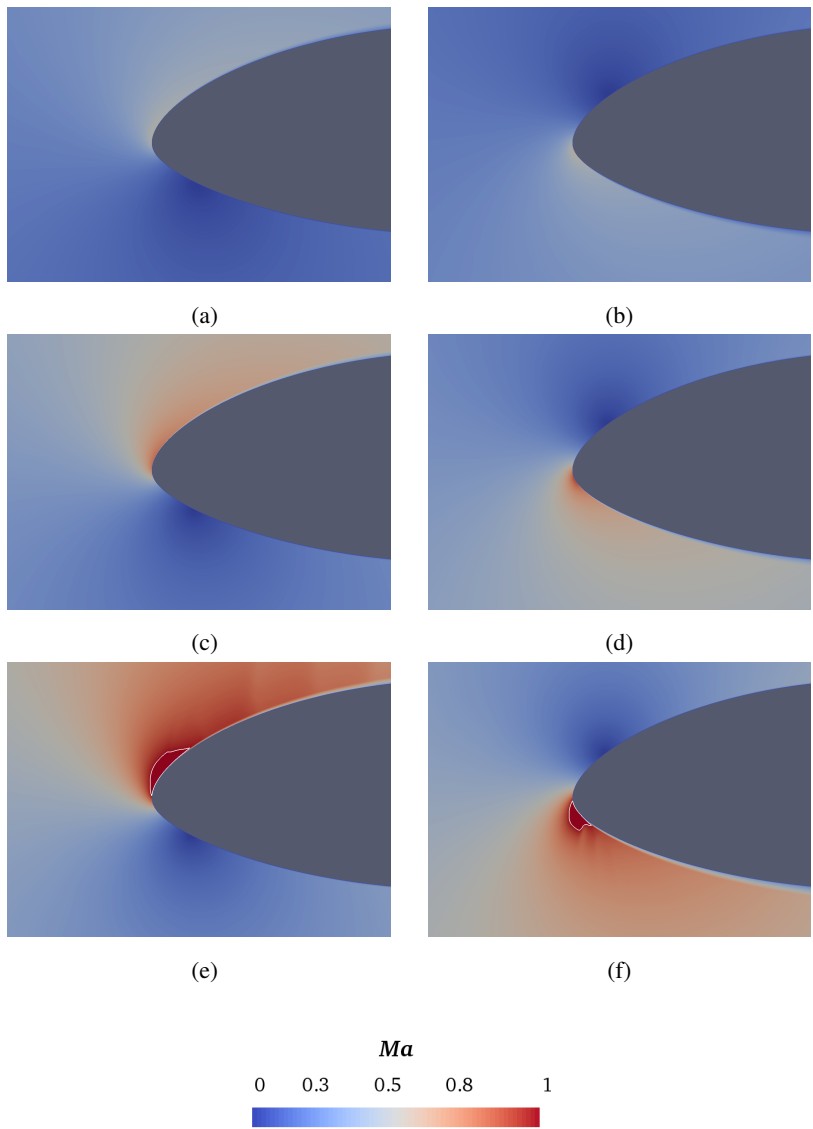

**Figure 5.** Instantaneous contour maps of local Mach number at Re $= 1.8 \times 10^6$, for varying free-stream velocity: $M_\infty = 0.2$ (top), $M_\infty = 0.3$ (middle), and $M_\infty = 0.4$ (bottom). Two different incidences are considered: $\alpha = 15°$ (left column) and $\alpha = -15°$ (right column). The iso-line corresponding to Ma $= 0.99$ is depicted.

the Mach number reaches a local maximum of 1.48 (see Figure 6 (f)), with the flow field being characterized by the presence of a single shock wave (see Figure 8 (f)), downstream of which the flow reattaches in the subsonic regime.





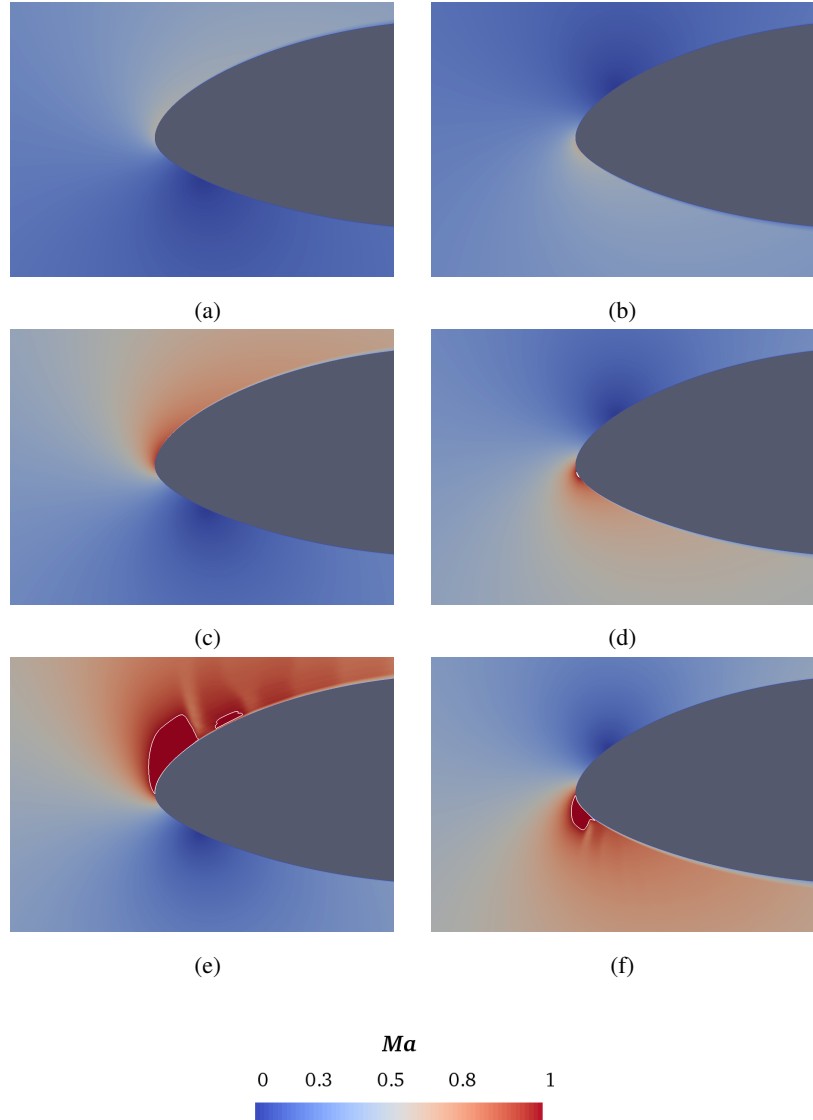

**Figure 6.** Instantaneous contour maps of local Mach number at Re $= 9 \times 10^6$, for varying free-stream velocity: $M_\infty = 0.2$ (top), $M_\infty = 0.3$ (middle), and $M_\infty = 0.4$ (bottom). Twoo different incidences are considered: $\alpha = 15°$ (left column) and $\alpha = -15°$ (right column). The iso-line corresponding to Ma=0.99 is depicted.

### 3.1.2 Quantitative analysis

The quantitative analysis is performed by examining the mean pressure coefficient distribution on the airfoil. Figures 9 and 10
show these data for all the different flow scenarios analyzed in the present work. The minimum suction peak is compared with
the critical value at which the flow over the airfoil is locally attaining a supersonic value. In these plots, the critical pressure



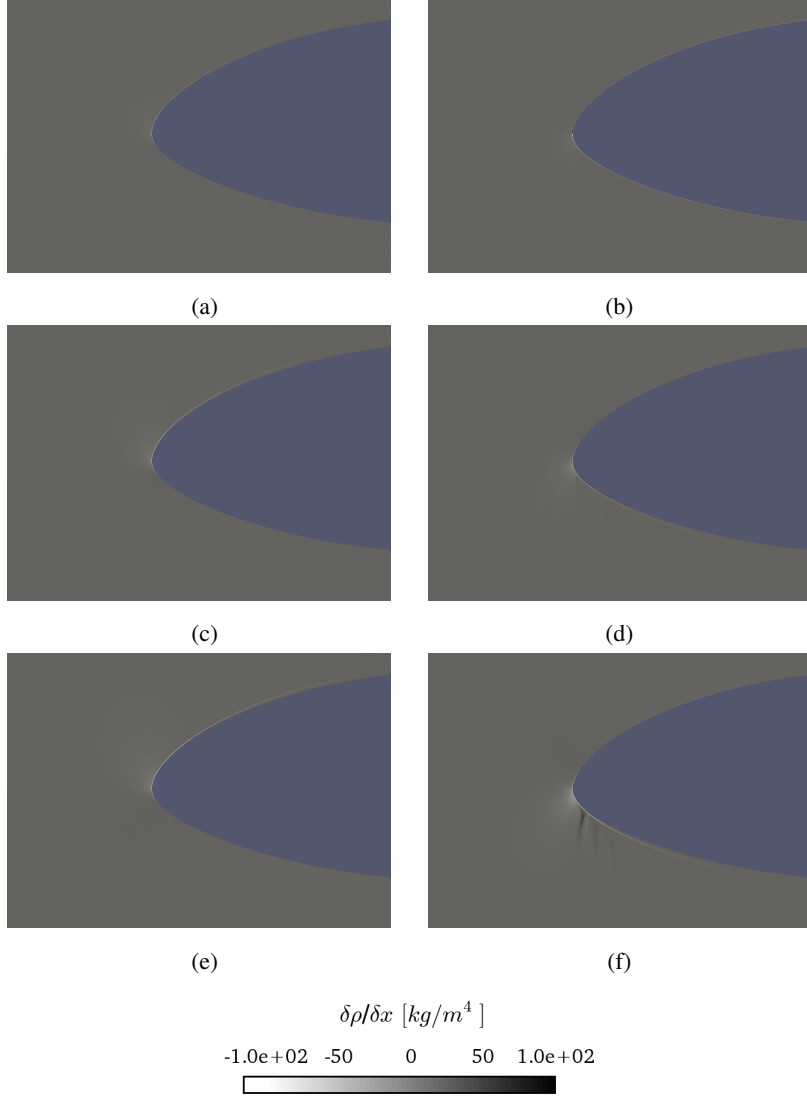

$\delta\rho/\delta x \; [kg/m^4]$

-1.0e+02    -50    0    50    1.0e+02

**Figure 7.** Instantaneous (numerical) Schlieren images at Re = $1.8 \times 10^6$, for varying free-stream velocity: $M_\infty$ = 0.2 (top), $M_\infty$ = 0.3 (middle), and $M_\infty$ = 0.4 (bottom). Two different incidences are considered: $\alpha = 15°$ (left column) and $\alpha = -15°$ (right column)

.

coefficient is expressed by a red solid line, corresponding to the values of -16.31, -6.95, and -3.66, for $M_\infty = 0.2$, 0.3, and 0.4, respectively. Note that this line is not reported in Figures 9 (a) and 10 (a), being out of the represented range. A local supersonic flow is established in the region where the pressure coefficient overcomes the corresponding critical value, as is shown in Figures 9 (c) and 10 (c). Note that, in Figure 10 (b), the minimum suction peak for the negative incidence is almost equal to the critical value, which makes the presence of local supersonic flow questionable. For the positive incidence, the


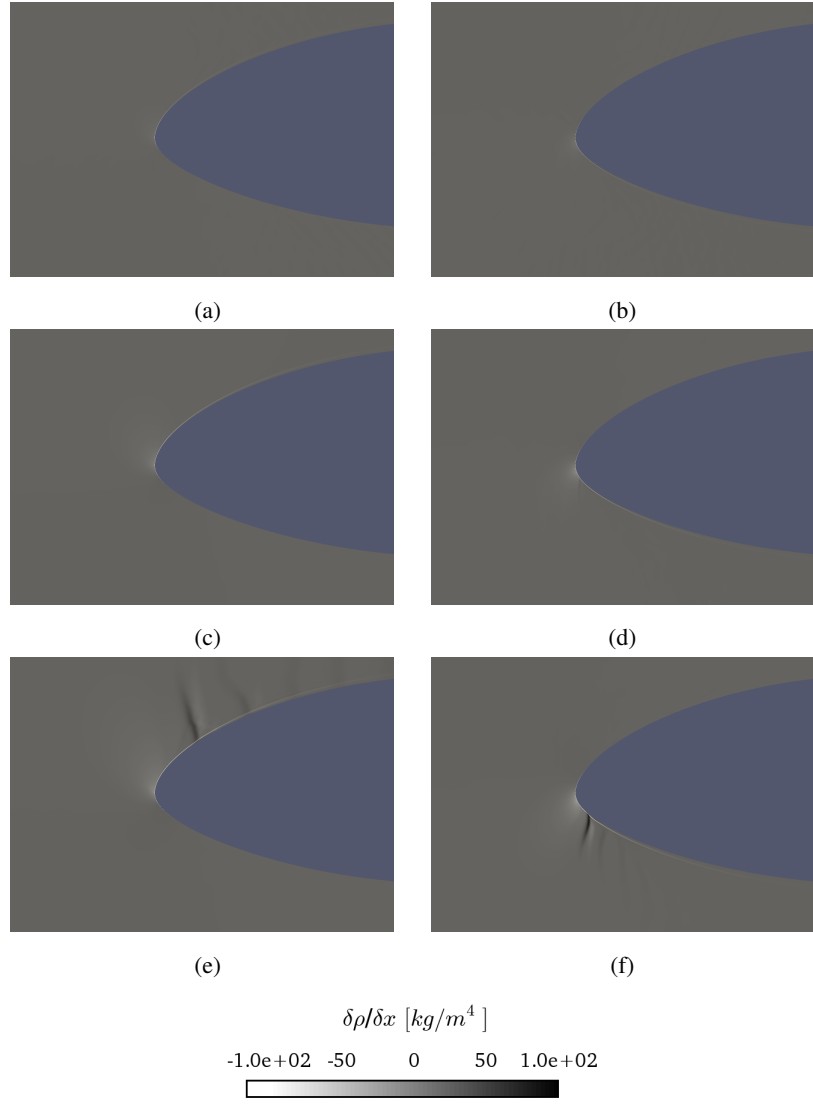

**Figure 8.** Instantaneous (numerical) Schlieren images at Re $= 9 \times 10^6$, for varying free-stream velocity: $M_\infty = 0.2$ (top), $M_\infty = 0.3$ (middle), and $M_\infty = 0.4$ (bottom). Two different incidences are considered: $\alpha = 15°$ (left column) and $\alpha = -15°$ (right column)

.

maximum suction peak stabilizes at approximately 4.2, whereas for the negative incidence, it escalates up to the value of 6. This result suggests the conclusion that the influence of the Mach number is notably more pronounced at higher Reynolds numbers.

Figures 11 and 12 show the instantaneous skin friction coefficient distribution over the airfoil as a function of the camber, where [0,1] represents the lower camber and [1,2] the upper one. By inspecting these figures, it is possible to conclude that,





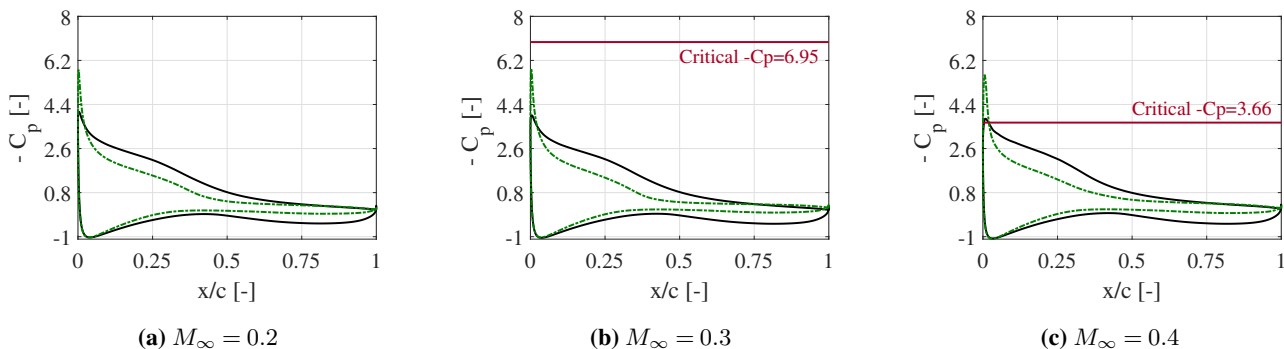

**(a)** $M_\infty = 0.2$      **(b)** $M_\infty = 0.3$      **(c)** $M_\infty = 0.4$

**Figure 9.** Mean pressure coefficient distribution at Re = $1.8 \times 10^6$, for $\alpha = 15°$ (black solid line), and $\alpha = -15°$ (green dashed line).

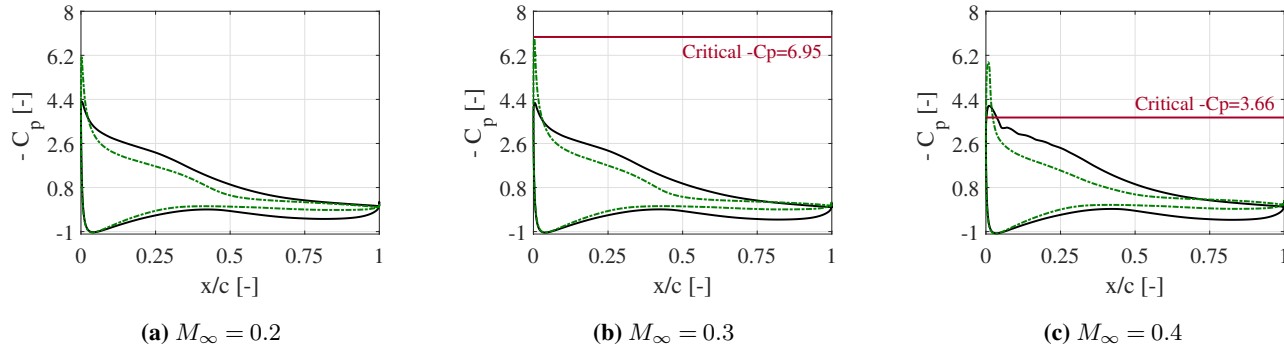

**(a)** $M_\infty = 0.2$      **(b)** $M_\infty = 0.3$      **(c)** $M_\infty = 0.4$

**Figure 10.** Mean pressure coefficient distribution at Re = $9 \times 10^6$, for $\alpha = 15°$ (black solid line), and $\alpha = -15°$ (green dashed line).

with increasing $M_\infty$, the separated flow region becomes smaller, and the separation point moves towards the trailing edge. This analysis also confirms that increasing the free-stream velocity parameter leads to a separation delay. It is also interesting to note the presence of discontinuities in the skin friction coefficient distributions for the flow field configurations where transonic flow exists. These discontinuities can certainly be attributed to the presence of non-linear phenomena, such as shock waves.

### 3.2 Comparison with compressible correction

The substantial difference between the present work and the previous one by De Tavernier and von Terzi (2022) lies in the current utilization of the URANS approach. Figure 13 illustrates the critical combination of angles of attack and inflow Mach numbers, indicating the threshold above which localized supersonic flow occurs on the FFA-W3-211 tip airfoil. Results obtained from URANS and Xfoil, using the Prandtl-Glauert compressibility correction for fully turbulent flow, are presented for different Reynolds numbers. Notably, at the same Mach number, URANS predicts the onset of transonic flow at slightly smaller angles of attack. In particular, when the wind turbine operates beyond its rated wind speed (with negative angles of attack), URANS suggests a larger safety margin. On the other hand, if the turbine operates below its rated wind speed (with positive angles of attack), the inverse relationship holds true. Furthermore, this analysis highlights that, for both URANS





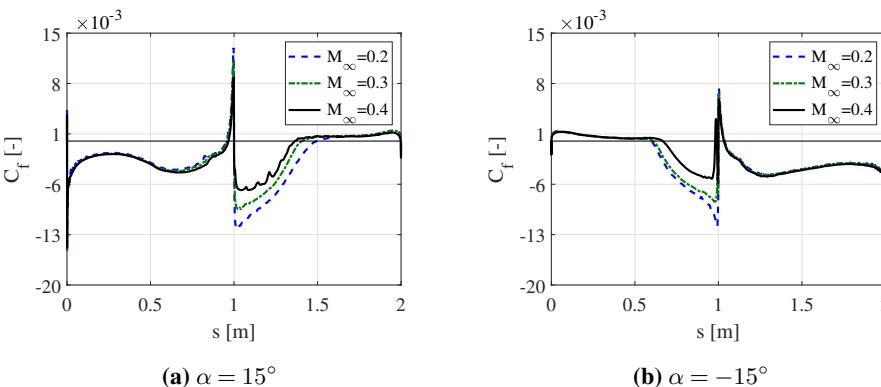

**Figure 11.** Instantaneous skin friction coefficient distribution at Re $= 1.8 \times 10^6$, for the two different incidences, and varying inflow Mach number

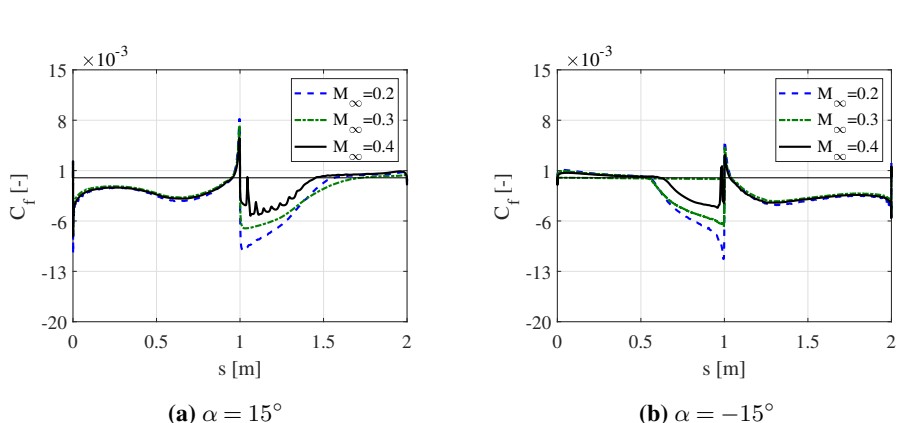

**Figure 12.** Instantaneous skin friction coefficient distribution at Re $= 9 \times 10^6$, for the two different incidences, and varying inflow Mach number

and Xfoil data, an increase in the Reynolds number promotes the transition to local supersonic flow conditions, especially for larger (either negative or positive) angles of attack. As a result, conventional wind tunnel studies, typically conducted at Reynolds numbers lower than those experienced by the wind turbine during operation, may provide less conservative safety zone predictions. Despite the overall agreement between the two techniques, some discrepancies are observed. In particular, a certain difference is apparent around the Mach 0.6 peak, which can be attributed to insufficient data analyzed in this study.

Another discrepancy is observed at angles of attack over $12°$, for Re $= 1.8 \times 10^6$. This latter mismatch resides in the post-stall region, as was demonstrated by Bertagnolio et al. (2001), where both Xfoil and URANS approaches fail. Nevertheless, it is important to highlight that both operational scenarios deviate significantly from the conditions faced by wind turbines that are at risk of experiencing transonic flow.



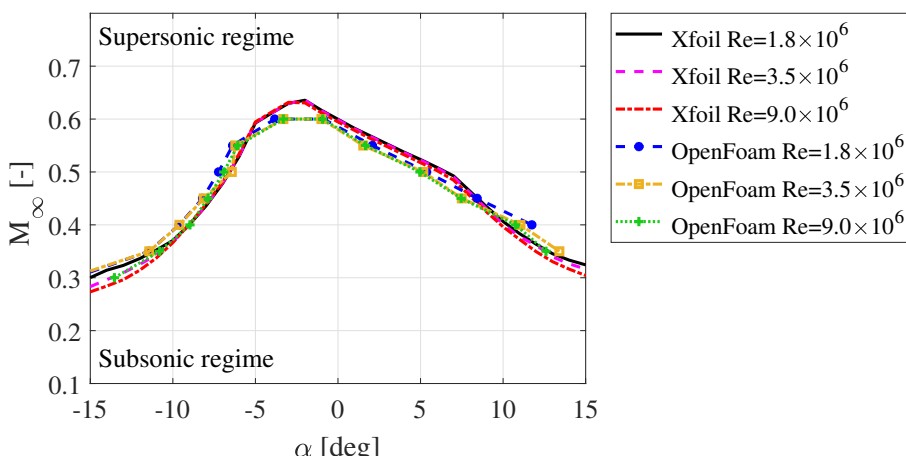

**Figure 13.** Subsonic-supersonic boundary for the FFA-W3-211 wind turbine tip airfoil: Xfoil $\text{Re} = 1.8 \times 10^6$ (black line), Xfoil $\text{Re} = 3.5 \times 10^6$ (magenta line), Xfoil $\text{Re} = 9 \times 10^6$ (red line), OpenFoam $\text{Re} = 1.8 \times 10^6$ (blue line), OpenFoam $\text{Re} = 3.5 \times 10^6$ (yellow line) and OpenFoam $\text{Re} = 9 \times 10^6$ (green line).

### 3.3 Shock waves detection

In this section, the potential presence of shock waves in transonic regime is analyzed. The study is necessary since the appearance of shock waves gives rise to severe pressure changes that can adversely affect the wind turbine's performance and loading.

Figure 14 shows the various configurations, in terms of angle of attack and inflow Mach number, that are selected for obtaining the URANS threshold in the previous analysis, as they are represented by the grey symbols. For each configuration,

the red symbols indicate the appearance of (local) supersonic flow. For these latter supersonic flow conditions, the presence of shocks is detected using the mathematical relation in Equation (4), and represented here by the green symbols in the figure.

It is noteworthy that a substantial dependency lies in the inflow Mach number. In particular, an increase in this parameter results in the emergence of shock waves, even at modest attack angles. Conversely, at low inflow Mach numbers, shock waves do not arise even at very high attack angles. Apparently, the presence of a local supersonic flow is not sufficient for the shock

waves to appear. In fact, in some circumstances, the acceleration of the flow is sufficiently gradual to prevent the formation of a discontinuity for the thermodynamic variables, confirming the results illustrated in Figure 5 (e) 7 (e). This analysis is performed for three different Reynolds numbers: $1.8 \times 10^6$, $3.5 \times 10^6$, and $9 \times 10^6$. Figure 14 shows a marked dependence of the solution on the Reynolds number, since increasing this parameter promotes the appearance of transonic flow, as well as the increment of configurations for which shocks appear in the flow field.





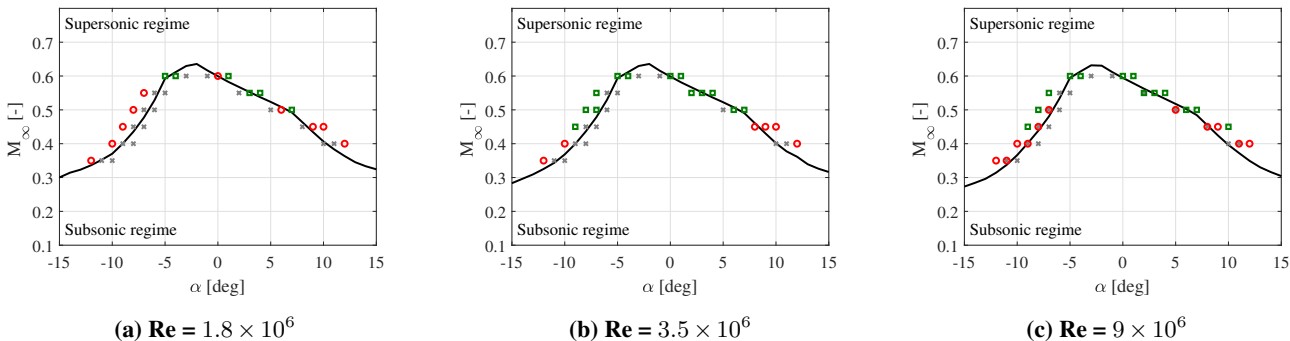

(a) **Re** = $1.8 \times 10^6$  (b) **Re** = $3.5 \times 10^6$  (c) **Re** = $9 \times 10^6$

**Figure 14.** Subsonic-supersonic boundary for the FFA-W3-211 wind turbine tip airfoil (using Xfoil): configurations selected for URANS threshold (grey crosses), configurations in which a supersonic regime is established (red circles), and configurations in which shock waves appear (green square).

## 4 Conclusions

This paper investigated the aerodynamic features of the FFA-W3-211 wind turbine tip airfoil under compressible and unsteady flow conditions. For this purpose, a URANS approach was described and validated with experimental data from the literature. The presented URANS set-up predicted the mean turbulent flow around the airfoil and was further used to analyze the impact of the Mach number and Reynolds number.

It was found that strong compressibility effects need to be taken into account to accurately predict the aerodynamic performance and loads of large wind turbine rotors in realistic operational conditions. Moreover, it was shown that transonic flow appears in some circumstances. This is the case particularly at high negative angles of attack, which correspond to the incidences encountered by the tip section of the blade at the high wind speeds encountered in above-rated wind conditions close to cut-out.

The threshold between subsonic and supersonic flow, using a fully compressible URANS formulation, was determined. This analysis was performed for Reynolds numbers varying from a characteristic value used in wind tunnel experiments to one representative of the IEA-15MW RWT rotor. A marked dependence on the Reynolds number, especially for high incidences, was observed. It was found that an increase in the Reynolds number promotes the onset of local supersonic flow conditions. The presence of local supersonic flow, however, was shown to be an insufficient criterion for shock waves to appear.

These results corroborated conclusions previously found using compressibility corrections. However, at high incidences, for the same angle of attack, the new URANS results showed a higher incoming velocity for which a local supersonic regime was established compared to the Xfoil predictions with compressibility corrections. This is very important as it suggests that, taking the transonic threshold as the design limit, Xfoil would predict a lower operational range resulting in a more conservative approach than URANS calculations.

It is worthwhile to recall the limitations of URANS in accurately capturing transonic flow phenomena due to inherent assumptions in the turbulence modelling and the associated interaction with shocks. In fact, the Reynolds-averaging procedure





inherently either eliminates or, in case of strong flow instabilities, attenuates a significant amount of flow unsteadiness while shifting the dominant frequency towards lower values (Fröhlich and von Terzi , 2008). It is also important to note that only fully turbulent flow conditions were considered in this study, whereas laminar to turbulent transition could occurs on clean wind turbine airfoils. Note that, therefore, for consistency with the fully turbulent flow of the URANS, compressibility correction analyses presented here were repeated with tripped conditions rather than the natural transition used in De Tavernier and von Terzi (2022).

The presence of supersonic flow raises research questions regarding the impact of shock waves and buffeting on wind turbine performance and lifetime. Indeed, it will be crucial to assess these effects to ensure efficient operation and durability of the next-generation large wind turbines. Due to the highly unsteady and three-dimensional nature of the phenomenon and the need to predict dominant frequencies in the flow field to assess aeroelastic instabilities, it is recommended to conduct experiments and/or high-fidelity simulations. Along this line of research, due to the limitations associated with the high flow Reynolds number in transonic wind tunnels, future investigations may want to consider employing Large Eddy Simulations (LES) with advanced wall models or hybrid RANS/LES techniques, like the ones recently proposed by Salomone et al. (2023).

*Code and data availability.* The code and data can be provided on request by contacting M. Cristina Vitulano

*Author contributions.* MCV conducted the overall research under the supervision of DDT, GD and DvT. The conceptualisation of the research and the methodology was theorized by MCV, DDT and DvT. The implementation of the computational model and the investigations were carried out by MCV. The results were analyzed and visualised by MCV, DDT, GD and DvT. MCV wrote the original draft, which was reviewed and edited by DDT, GD and DvT.

*Competing interests.* The authors declare that they have no conflict of interest



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
