# Peer review of "Numerical Analysis of Transonic Flow over the FFA-W3-211 Wind Turbine Tip Airfoil"

_Wind Energy Science, 2024_

## Author Comment (AC1)

**Response to the Referee report # 1**

Dear Referee,

Thank you for the time and effort you dedicated to providing feedback on our manuscript. We are grateful for your insightful comments and valuable suggestions that helped us to improve our paper. Please find below, highlighted in blue, our point-by-point response to your comments and concerns (in red). All page numbers refer to the revised version of the manuscript with tracked changes.

This article provides an investigation of the effects of compressibility on 2D airfoil flows in the context of wind turbine blade application at high Reynolds numbers and realistic Mach numbers for modern wind turbines.

The manuscript is clearly and well-written. The methodology and scientific content appear correct and error-free. Only in a few places, the reviewer would like some clarifications and small corrections, as reported below.

• **Comment 1**: Introduction, 1.35-40: It is reported that 1) Yan and Archer found that neglecting compressibility could result in wind farm power production overestimation, 2) Campobasso et al and Ortoloni et al found that compressibility produces an increase of peak rotor power. This appears contradictory, and the authors should comment here if there are different mechanisms at stake, or at least point out this contradiction.

**Response**: Thank you for highlighting this issue. We agree with your comment. The main reason for this contradiction is that Yan and Archer used a BEM-based method with compressibility extension while Campobasso et al. and the other cited literature used CFD. As Sørensen et al. showed, this can lead to differences, where CFD had shown better agreement with experiments. This is the reason why, in this study, both approaches were compared. To make this more clear we rearranged the corresponding paragraph in the introduction [lines 28-40].

• **Comment 2**: Section 2.2, l. 133: The authors refer to "dynamic stall". There seems to be some confusion here. The present calculations are done at constant angle of attack, and the authors probably means "stall" or "separation". Dynamic stall refers to the dynamic behavior of airfoil separation and stall when the angle of attack is varying.

**Response**: Thank you for pointing this out. Indeed, we meant "stall", and the point has been corrected in the revised manuscript.

• Comment 3: Fig 6, caption: "Twoo"

Response: Thank you for bringing this to our attention. It has been corrected.

• **Comment 4**: Section 3.3: The interpretation of Fig. 14 needs clarification. If the reviewer understands well, all symbols are URANS calculations. When a supersonic region is present, the grey symbols are replaced by red ones, which are in turn replaced by green ones when

a shock is detected. It is not clear why the authors did not conduct calculations for an angle of attack of 2 degs, M=0.6 and 0.65, for all Reynolds numbers, in order to refine the threshold curve shapes in this region, as illustrated by the horizontal line in Fig. 13 in this region.

**Response**: Thank you for your feedback. We have included a clarification about Fig. 14 in lines 249-252 accordingly. Additionally, we decided not to carry out more detailed analyses beyond an inflow Mach number of 0.6, as this area is not of major importance as it will not occur during wind turbine operation. This decision has been explicitly stated in the final manuscript on lines 235-238

• **Comment 5**: Furthermore, it is noticed (l.244-245) that increasing Reynolds number promotes the appearance of transonic flow. Couldn't it be surmised here that this is caused by the decreasing viscosity? Or could there be another explanation?

**Response**: You have raised an important point here that we did not elaborate on in detail in the manuscript. Please note that, in our CFD-based study, we were able to vary Mach and Reynolds number independently, whereas, in experiments, this is very difficult and typically not achieved. Hence, keeping both dimensionless parameters the same in the wind tunnel as on a real wind turbine is quite a challenge and the use of CFD to relate both results may be needed. For the same Mach number, but different Reynolds numbers, compressibility of the fluid is kept constant but the dimensionless thickness of the boundary layer on the airfoil changes and hence the streamlines around the airfoil. This could indeed be interpreted as a viscous effect. (Note that we excluded the effect of change in the laminar to turbulent transition by enforcing fully turbulent flow, i.e. the setup is equivalent to a tripped airfoil experiment or a dirty/rough wind turbine in the field).

• **Comment 6**: Finally, it could be mentioned somewhere in the manuscript that tip effects (where the compressibility effects are maximum) may also play a role in the scenarios presented in this paper.

**Response**: Thank you for your comment. This is indeed a valuable suggestion for future research, which we have described in lines 290-292 of the updated manuscript.

In conclusion, the paper is worth publishing and only minor revisions are suggested.

**Response to the Referee report # 2**

**Dear Referee,**

Thank you for the time and effort you dedicated to providing feedback on our manuscript. We are grateful for your insightful comments and valuable suggestions that helped us to improve our paper. Please find below, highlighted in blue, our point-by-point response to your comments and concerns (in red). All page numbers refer to the revised version of the manuscript with tracked changes.

The article addresses the possibility of transonic/supersonic flow occurrence in the blade tip region of large wind turbines. The manuscript is well organized and well written, and relevant to the field. An interesting cross comparison of high- and low-fidelity aerodynamic methods to foresee the possible occurrence of transonic/supersonic flow is also presented.

Some additional comments are as follows. I would recommend the authors addressing them in the revised manuscript.

• **Comment 1**: Mesh refinement should be addressed, indicating if or providing evidence that the presented parametric analyses provide mesh-independent results.

**Response**: Thank you for raising this point. By inspection of Figure 4, where the URANS solutions for three different mesh resolutions are reported, it is possible to verify that the numerical solution converges to the experimental data (taken as a reference for the validation of the numerical model). Moreover, the CFD results remained constant when the resolution of the mesh was changed. A statement regarding the mesh-grid independence assessment has been explicitly addressed in the final manuscript [lines 141-143].

• **Comment 2**: Could authors indicate the turbulence intensity of the results in Fig. 1? I would imagine the probability of transonic flow occurrence increases with the turbulence intensity, that tends to be low offshore. This I would assume that the occurrence of transonic conditions is more likely onshore. Can authors comment on this in the paper?

**Response**: You have raised an important question. It is indeed true that an increase in turbulent intensity promotes the onset of transonic flow, and that the velocity variation is lower offshore. However, it is also true that inflow wind speed and blade dimensions play a key role in the formation of transonic flow, and both these parameters are higher offshore, given the higher average wind speeds and fewer constraints on blade size. Nevertheless, a detailed sensitivity analysis has been carried out in the previous work of De Tavernier and von Terzi. Here it is clearly shown that TI plays a significant role in the emergence of transonic flow conditions. This has been highlighted explicitly in the manuscript (line 56). By the way, we have specified the class of wind in the caption of Figure 1.

• **Comment 3**: Page 4.Article reads: 'The aim of the first part of this study is to prove that transonic flow can appear even in normal operation conditions'. Please define 'normal operating conditions'.

**Response**: Thank you for your comment. By "normal operational condition," we refer to conditions as defined in the IEC design standard for wind turbines (IEC61400-1). For additional details, please refer to the previous study of De Tavernier and von Terzi, 2022. Following the suggestion, this has been added in the revised manuscript (lines 94-97).

• Comment 4: Page 4, article reads: 'Also, to save computational resources, wall functions are used to model the boundary layer region'. AoAs of ± 15 degrees are quite high. Is there not the chance that shock-induced separation due to boundary layer/shock interactions may occur? In this circumstance, use of wall functions may induce significant errors. Can authors please comment on this matter in regard to the presented results?

**Response**: You have raised an important point here. It is true that the use of wall functions adds another uncertainty to the results for some flow conditions. However, this work aims to evaluate the conditions under which the effects of transonic flow need to be considered. The objective, therefore, is to analyze as many scenarios as possible while maintaining an acceptable level of reliability (achieved through a CFD-based approach) and minimizing computational costs (hence the choice to use wall functions). Here, we compared URANS with compressibility-corrected Xfoil calculations. The working assumption is that, as long as both methods agree, there is a relatively high confidence in the results. Once they depart, future analyses using high-fidelity models are necessary to confirm the findings. This is now stressed in the validation section [lines 114-115] and in the conclusions [lines 284-287]. We also references the adaptive wall-function used in the methodology section [lines 114-115].

• Comment 5: Fig. 3, Table 1. This type of analysis should be performed with the AoA of ± 15 degrees, since pressure perturbations on either airfoil size are more likely to extend farther away from the airfoil, potentially leading to spurious reflections. I am not asking to redo this analysis, but some comments on the choice of 7.99 degrees may be helpful.

**Response**: Thank you for highlighting this point. The choice of 7.99° lies in the intention to perform a preliminary validation by focusing on a single factor: the domain size. Indeed, at  $\alpha = 7.99^{\circ}$ , the flow is still attached, in the linear part of the polar, and experimental and numerical results should match (also seen in the reference of Bertagnolio, 2001). So, by changing the size of the domain, any loss of this match would almost certainly be due to the change in the domain size. Instead, at an incidence of 14.98°, this "form of control" is lost. At  $\alpha = 14.98^{\circ}$  the flow is separated and stall is approaching, causing a mismatch between numerical and experimental results. Therefore, a further discrepancy does not necessarily lie in the variation in the domain size but could depend on other factors. Nonetheless, further validation is conducted for  $\alpha = 14.98^{\circ}$ , considering only the smallest domain in Figure 4c. However, this point has been clarified in the revised manuscript [lines 132-135].

• **Comment 6**: Fig. 4c: is it not possible that additional reason for the discrepancy Open-FOAM/experiments could also be the use of wall functions in separated flow region?

**Response**: Yes, we agree. This point is now stressed in the revised manuscript on lines 145-146

• **Comment 7**: Perhaps figures 11 and 12 could be slightly improved? For example, indicating LE is at s=1 (I suppose). Indication of upper and lower side in cf graphs may also help quicker readability.

**Response**: Thank you for your suggestion, Figures 11 and 12 have been improved in the final manuscript.

---

## Author Response (AR2)

**Response to the Referee report # 3**

Dear Referee,

Thank you for the time and effort you dedicated to providing feedback on our manuscript. We are grateful for your insightful comments and valuable suggestions that helped us to improve our paper. Please find below, highlighted in blue, our point-by-point response to your comments and concerns (in red). All page numbers refer to the revised version of the manuscript with tracked changes.

I would like to thank the authors for the answers to the remarks and questions of the first review and the revision of the manuscript!

In my opinion, the chosen domain size of 10 chord lengths is not sufficient for investiga-tions in the stall range and I suspect that at alpha = +/-15 deg. a dependence of the re-sults on the domain size cannot be excluded. The authors should at least reconsider the setup for future studies.

Nevertheless I can agree to the publication of the contribution, but would be pleased if the authors could take the following minor remarks into account in the final version:

- **Comment 1**: Line 217 of the revised manuscript: Also to be consistent with the wording on line 224, "maximum suction peak" should be written here and not "minimum suction peak".

  **Response**: Thank you for your comment. It has been corrected in the final manuscript (line 214).

- **Comment 2**: Line 227, dto. Figs. 11 and 12: This probably refers to the arc length and not to the camber of the airfoil. I would suggest: "lower camber" - "lower side", dto. in Figs. 11 and 12.

  **Response**: Thanks for pointing this out. This review has been implemented in the final manuscript (lines 217-218 and fig. 11 and 12 as well).

- **Comment 3**: Figs. 11 and 12: Obviously -cf and not cf is shown here. This should be clarified.

  **Response**: Thank you for bringing this to my attention. It has been corrected in the final manuscript.

- **Comment 4**: Line 237: According to my information, standard XFOIL contains the Karman-Tsien and not the Prandtl compressibility correction.

  **Response**: Thank you for raising this point. You are correct that XFOIL utilizes the Karman-Tsien compressible correction; however, we have implemented the Prandtl-Glauert correction.